# Emerging TACnology: Heterobifunctional Small Molecule Inducers of Targeted Posttranslational Protein Modifications

**DOI:** 10.3390/molecules28020690

**Published:** 2023-01-10

**Authors:** Pascal Heitel

**Affiliations:** Chemistry Research Laboratory, Department of Chemistry, University of Oxford, Mansfield Road, Oxford OX1 3TA, UK; heitel@pharmchem.uni-frankfurt.de

**Keywords:** heterobifunctional molecules, proximity-inducing chimeras, targeted protein modification, conditional protein degradation

## Abstract

Posttranslational modifications (PTMs) play an important role in cell signaling and they are often deregulated in disease. This review addresses recent advances in the development of heterobifunctional small molecules that enable targeting or hijacking PTMs. This emerging field is spearheaded by proteolysis-targeting chimeras (PROTACs), that induce ubiquitination of their targets and, thus, tag them for degradation by the proteasome. Within the last decade, several improvements have been made to enhance spatiotemporal control of PROTAC-induced degradation as well as cell permeability. Inspired by the success story of PROTACs, additional concepts based on chimeric small molecules have emerged such as phosphatase-recruiting chimeras (PhoRCs). Herein, an overview of strategies causing (de-)phosphorylation, deubiquitination as well as acetylation is provided, and the opportunities and challenges of heterobifunctional molecules for drug discovery are highlighted. Although significant progress has been achieved, a plethora of PTMs have not yet been covered and PTM-inducing chimeras will be helpful tools for chemical biology and could even find application in pharmacotherapy.

## 1. Introduction

Genetic information is encoded by DNA and determines heritable traits. However, phenotypic characteristics are not exclusively influenced by the underlying genome and the field of epigenetics studies phenotypic changes which cannot be explained by alterations in the DNA sequence. For instance, caterpillars and butterflies share the same genotype but express distinct phenotypes. On the molecular level, this can be explained by the differential transcription of genes into RNA and the translation of RNA into proteins. Further diversification of the proteins can occur posttranslationally, giving rise to broad phenotypic diversity. Posttranslational modifications (PTMs) are present almost universally across the proteome and generally cause small yet significant changes in protein structure and function (Figure 1) [1]. These can have vast consequences for the regulation of enzymatic activity, protein localization, protein folding, and protein-protein interactions (PPIs) [1]. To date, more than 450 PTM types are listed in the UniProt database [2] and many proteins contain more than one PTM, thus further increasing the combinatorial space of modified proteins [1]. This highly versatile platform allows rapid adjustment to environmental and physiological influences that cannot be addressed by the genome or epigenome on the same time scale, and besides, the genome is much smaller than the proteome [3]. The most frequently occurring PTM is phosphorylation, followed by glycosylation, acetylation, ubiquitination, methylation, and covalent attachment of small ubiquitin-like modifier (SUMO) proteins called SUMOylation [4]. Due to the high occurrence and importance of PTMs, alterations in a protein’s PTM profile are often associated with diseases [5]. Hence, chemical tools to precisely manipulate the PTM profile of proteins are a valuable approach to modify protein properties including enzymatic activity and interactions with other proteins and could achieve a therapeutic effect.

This review addresses heterobifunctional small (and medium-sized molecules) which alter the PTM profile of a specific protein of interest (POI). In principle, such compounds could constitute a new class of drugs that gives access to undruggable proteins and diseases. Here, the opportunities and challenges of these small and medium-sized heterobifunctional molecules for drug discovery are also highlighted. To date, the large majority of chimeric molecules cause ubiquitination of their target protein and subject it to proteasomal degradation, but recent developments transfer the original concept to phosphorylation and acetylation (ubiquitin-like modifier (SUMO) proteins called SUMOylation [4]. Due to the high occurrence and importance of PTMs, alterations in a protein’s PTM profile are often associated with diseases [5]. Hence, chemical tools to precisely manipulate the PTM profile of proteins are a valuable approach to modify protein properties including enzymatic activity and interactions with other proteins and could achieve a therapeutic effect. Figure 1 and Figure 2). Most PTMs have not yet been targeted by small molecule chimeras.

## 2. Ubiquitination and Autophagy

Ubiquitin is a small, 76-amino acid regulatory protein which got its name from the ubiquitous expression [6]. It can be attached to proteins either at the ε-amino group of lysine or at the *N*-terminus. Ubiquitination requires a multi-step process involving three enzyme classes: E1 ubiquitin-activating, E2 ubiquitin-conjugating, and E3 ubiquitin-ligase (Figure 3). First, the ubiquitin *C*-terminus forms a thioester bond with a cysteine residue of an E1 enzyme in an ATP-dependent manner. Then, an E2 enzyme mediates the transfer of the ubiquitin molecule from E1 to a cysteine residue in its active site via transthioesterification reaction. Last, an E3 enzyme interacts with both the conjugated E2 protein and a POI to catalyze the ubiquitination of the target [6]. Ubiquitin, itself, can be ubiquitinated in an iterative process (at one of its six lysine residues or the *N*-terminus), resulting in a polyubiquitin chain. Depending on the number of ubiquitin molecules and their linkage, ubiquitination signals differently affect protein localization, activity or stability. Frequently, E3 complexes tag their substrate with a K48-linked polyubiquitin chain, targeting the POI for 26S proteasomal degradation [7], which can be exploited by proteolysis-targeting chimeras (PROTACs).

### 2.1. PROTACs

PROTACs are heterobifunctional molecules comprising ligands for the POI and an E3 ligase which are connected via a linker moiety (Figure 4). They have been extensively reviewed elsewhere [8,9,10]. There are more than 600 E3 ligases known in human [7], of which cereblon (CRBN), cellular inhibitor of apoptosis (cIAP) 1, von Hippel-Lindau (VHL), and mouse double minute 2 homolog (MDM2) are commonly employed for PROTACs [11,12]. Historically, the use of CRBN as an E3 ligase developed from immunomodulatory imide drugs (IMiDs) such as thalidomide, lenalidomide, and pomalidomide that exhibit their anti-cancer and immunosuppressant activity by binding the CRBN subunit of the CRL4A^CRBN^ E3 ligase complex. The IMiD binding mode modifies the molecular surface, directly enabling new protein-protein interactions so that neosubstrates can be recruited including Ikaros (IKZF1), Aiolos (IKZF3), CK1α, and GSPT1 [12,13]. As a result, the neosubstrates are polyubiquitinated and subsequently subjected to degradation by the proteasome. In contrast to PROTACs which require the presence of two distinct ligands for POI and substrate, IMiDs can be characterized as monovalent inducers of PPIs and thus are so-called molecular glues. Thalidomide and its derivatives have been exploited in the development of several PROTACs [11,14].

PROTACs hijack the proteasome-dependent protein degradation pathway. An optimal linker enables a conformation to form a stable POI:PROTAC:E3 ligase ternary complex [15,16]. Ubiquitin is transferred multiple times to POI lysine through the E3 complex and the proteasome recognizes this motif of degradation (Figure 4). The PROTAC, itself, is not degraded during this process and is free to undergo another catalytic cycle, provided that the chimera does not bind irreversibly. Remarkably, the PROTAC does not necessarily need to target the canonical binding pocket. Thus, PROTAC technology offers access to the so-called undruggable proteome [8]. To date, only 754 proteins are targeted by FDA-approved drugs [17]. The principal reason for this is that 80% of proteins lack a binding site that regulates protein function [18]. For a PROTAC, it is not necessary to have a ligand that binds to the canonical binding pocket. In principle, any allosteric site can be targeted to generate a functional degrader. Furthermore, cellular concentrations do not need to be high because PROTACs and glues act catalytically. PROTACs hold the potential to be a breakthrough innovation to drug discovery, albeit no PROTAC has gained market approval so far.
Figure 4(**a**) Proteolysis-targeting chimeras (PROTACs) are heterobifunctional molecules which bind to the protein of interest (POI) as well as an E3 ligase leading to polyubiquitination and subsequent proteasomal degradation. Provided that the PROTAC bond is reversible, they act as a catalyst. (**b**) Exemplary PROTAC targeting BRD4 as POI and VHL as an E3 ligase [16].
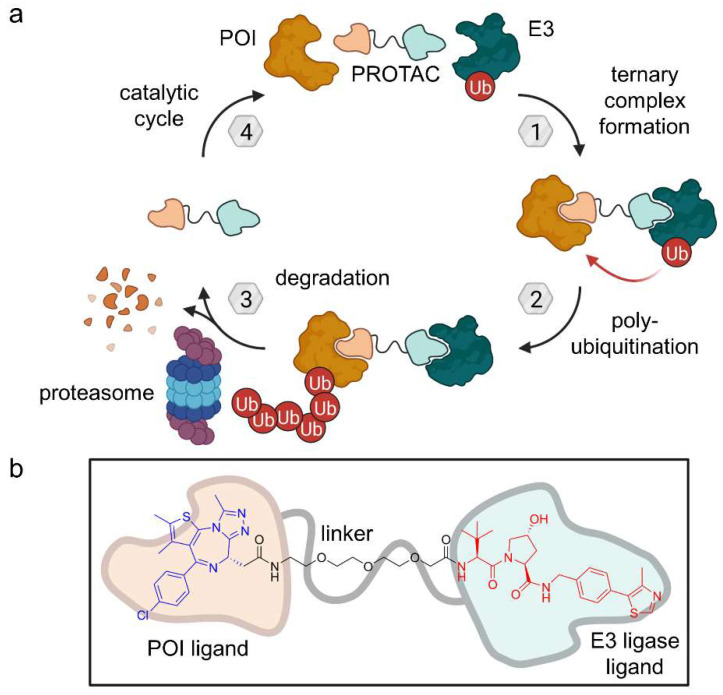


The therapeutic application of PROTACs is currently limited by unfavorable pharmacological properties such as poor cell penetration, low bioavailability, poor solubility, and low stability caused by high molecular weight and often peptidic E3 ligase ligands [11]. These effects are a direct consequence of disobeying Lipinski’s Rule of Five [19]. Yet, oral availability is achievable because PROTACs do not behave like classical small molecules [20,21]. For instance in apolar solvents, they can fold to bury hydrophilic parts inside and expose hydrophobic parts on the outside [22]. Information on how to engineer pharmacokinetic properties of large PROTAC molecules is limited. Another challenge is the similar mode of action of PROTACs and molecular glues. As a matter of fact, some PROTACs might induce POI degradation via a molecular glue mechanism [23]. Future research will have to show how these mechanisms can be differentiated and whether it is possible to unite them and achieve synergistic effects.

For use in cancer therapy, PROTACs could display on-target toxicity if the corresponding POI is widely expressed [24]. Target degradation in unaffected tissue might cause serious adverse effects. However, this problem can be overcome by the choice of E3 ligase. For instance, an experimental BCL-X_L_ inhibitor for the treatment of B cell lymphoma exhibited on-target thrombocytopenia and did not obtain market approval [25]. Therefore, the inhibitor was turned into a PROTAC based on the E3 ligase VHL, which is poorly expressed in platelets and, consequently, the PROTAC did not induce thrombocytopenia while maintaining therapeutic efficacy against B cell lymphoma [25]. The limitations mentioned are not exclusive to PROTACs but also hold true for most heterobifunctional chimeras which are discussed below.

Recently, different strategies have been developed to face existing problems from PROTACs which will be discussed in the following.

#### 2.1.1. Light-Responsive PROTACs

In some cases, chemically induced protein knockdown is only desired in specific tissues or time-dependently. For this purpose, various PROTAC systems have been developed to allow spatiotemporal control of degradation. The field of photo-responsive PROTACs is in the most literal sense highly vibrant and has achieved excellent results. One strategy to control PROTAC function by light is to introduce a photoswitch based on an azo group in the linker. The *Z*- and *E*-isomer occupy different configurations and affect changes in spatial arrangement and total size of the PROTAC. Consequently, the isomers alter the three-dimensional structure of the ternary complex and, usually, one of the isomers loses its ability to induce POI degradation (Figure 5a). PROTAC activation or deactivation can be obtained by isomerization at different light wavelengths to enrich one species and is generally reversible.

The Crews and Carreira labs replaced the polyether linker in BET protein degrader ARV-771 by such a photoswitchable azo linker (Figure 6) [26]. By introducing neighboring *ortho*-difluorobenzene groups, they created a bistable azo system [26]. In *trans*-configuration (*E*-**1**), the so-called photoPROTAC is an active BET degrader in vitro, whereas the *cis*-configuration leads to a reduction of 3 Å in chain length and a concomitant complete loss of activity. Most likely, *Z*-**1** is unable to reach the second binding site while retaining favorable PPIs, but also the rigidity of the linker in contrast to flexible polyether linkers might contribute to the differential isomer activity (Figure 5a). Notably, the *trans*-photoPROTAC (*E*-**1**) achieved a BET selectivity that was not observed for the parent PROTAC ARV-771 [26].

Similarly, the Jiang and Trauner groups developed azo group-based light-responsive PROTACs for the tyrosine kinases ABL and BCR-ABL (**2**), BET family proteins BRD2-4 (**3**) and the prolyl *cis*-*trans* isomerase FKBP12 (compounds **4** and **5**), respectively [27,28]. There was no general propensity for one specific configuration to induce degradation. While **1** and **2** only showed activity in *E*-configuration, active degraders could be obtained from **3–5** as *Z*-isomers. In all of the cases, the azo group was incorporated as part of the linker, either centrally (**1**, **2**, **5**) or in the transition zone to CRBN ligands (**3**, **4**).

In addition to the photoswitch technology, there have been introduced photocleavable moieties into PROTACs (Figure 5b,c). These groups can be either introduced into the POI ligand or the E3 ligase ligand and physically block the PROTAC from binding. If irradiated with a defined wavelength, the light-responsive moiety is irreversibly cleaved and the degradation-inducing PROTAC is released, allowing the time and place of protein knockdown to be controlled. Almost simultaneously, three different labs reported the synthesis and characterization of photo-caged PROTACs (pc-PROTACs) [29,30,31]. Xue et al. [29] developed a photo-caged derivative (pc-PROTAC **6**) of dBET1, a known PROTAC for BRD4 [32]. They attached a 4,5-dimethoxy-2-nitrobenzyl (DMNB) group to the amide in the transition zone from POI ligand (+)-JQ1 to the linker (Figure 7). All other reported pc-PROTACs (**7–11**) have the photocleavable group on the E3 ligase ligand, either attached to the imide group of thalidomide (**7**, **9**–**11**) or the hydroxyproline of VHL ligand VH298 (**8**). Apart from BRD4 (**6**, **9**, **10**) [29,30,31], Bruton’s tyrosine kinase (BTK, **7**) [29], estrogen-related receptor α (ERRα, **8**) [30], and the anaplastic lymphoma kinase (ALK) fusion proteins EML4-ALK and NPM-ALK (**11**) have been degraded upon photolysis. In the absence of light, none of the caged PROTACs degraded their target protein. The photocleavable moieties employed were DMNB (**6**), the slightly larger 6-nitroveratryloxycarbonyl group (NVOC, **7**, **10**, **11**) or 6-nitropiperonyloxymethyl group (NPOM, **9**), and diethylaminocoumarin (DEACM, **8**). These groups were installed using amides (DMNB), imides (NVOC, NPOM), carbamates (NVOC), and carbonate esters (DEACM).

Remarkably, the photo-caged dBET1 derivative **10a** from the Deiters group [30] is very similar to the molecule Xue et al. [29] prepared (**10b**) but only **10a** proved to release a quantitative amount of dBET1 upon irradiation. The only difference between these molecules is the heteroatom in 4-position of thalidomide, which is either oxygen (**10a**) or nitrogen (**10b**). Both compounds were irradiated at 365 nm to release the active degrader, but it should be noted that the methods used for photolysis quantification are not comparable [29,31].

Systemic degradation of a specific protein can give rise to unwanted toxicity [33]. The introduction of photo-responsive groups offers a high potential for application of PROTACs in cancer to spare healthy cells and reduce potential off-target effects. The great advantage of this method is that it is non-invasive and, in addition, azo group-containing PROTACs are reversible. On the contrary, the photo-caged strategy is irreversible, and irradiation not only releases the PROTAC but also a side product in quantitative amount. Presently, both techniques require UV light, which has a low tissue penetration and is toxic to DNA in elevated doses [31]. Moreover, temporal control may be compromised by rapid isomerization of *E-* and *Z*-isomers of azo compounds in the absence of irradiation. These general limitations could be overcome by improving the spectral properties of photo-responsive groups to absorb in the near-IR region. In a very recent study, a caged BET-PROTAC was developed to be responsive to X-ray radiation [34], which has a high precision and deep tissue penetration. In a xenograft mouse model, X-ray radiation not only served as a stimulus to release the active PROTAC and control BET degradation spatiotemporally but also synergistically suppressed tumor growth [34]. However, it has yet to be demonstrated in vivo that such radiotherapy-triggered PROTACs can avoid systemic toxicity caused by off-tissue POI degradation.

#### 2.1.2. Hypoxia-Activated PROTACs

Besides light to activate PROTACs, other triggers can be employed to allow spatiotemporal control of degradation. PROTACs hold a great potential for the treatment of tumors, in which protein expression is often dysregulated. Hypoxia is a hallmark of tumor microenvironment, which can be used as a stimulus to activate PROTACs. The rapidly proliferating tumor cells have a high demand on oxygen that cannot be supplied by surrounding blood vessels. Cheng et al. developed PROTACs that can be activated by hypoxia to specifically degrade epidermal growth factor receptor (EGFR) in cancer cells and spare healthy tissue with normal EGFR expression and function [35]. Their hypoxia-activated PROTAC **12** is based on the marketed EGFR inhibitor gefitinib and lenalidomide as CRBN-targeting unit (Figure 8). These two ligands are connected by a common PEG chain and a 4-nitrobenzyl group is attached to the secondary amine in gefitinib as a hypoxia-activated leaving group. The nitro group can be reduced by nitroreductases in both normoxia and hypoxia. However, the resulting amine is rapidly reoxidized in normoxia.

Reoxidation does not occur in hypoxia due to lack of oxygen. In hypoxia, the amine readily undergoes fragmentation to release the active PROTAC. Indeed, a higher fraction of EGFR was degraded in HCC4006 cells after treatment with **12** in hypoxia compared to normoxia. While this method can confine target degradation to a specific tissue, it is questionable whether hypoxia-activated PROTACs can be administered orally because of the afore-mentioned poor blood supply in tumors.

#### 2.1.3. Click-Formed PROTACs (CLIPTACs)

Although small molecule-induced protein degradation is a valuable concept in drug discovery, current PROTACs are limited by their pharmacokinetic properties associated with their large molecular weight. Thus, PROTACs do not meet the criteria for Lipinski’s Rule of Five and have large polar surface areas leading to poor cell penetration [14,19,36]. In light of this, researchers at Astex Pharmaceuticals developed small molecule precursors that can form a PROTAC in live cells by a bio-orthogonal click reaction [36]. These so-called click-formed PROTACs (CLIPTACs) consist of two components: a tetrazine-tagged thalidomide derivative (Tz-thalidomide) and a *trans*-cyclooctene-functionalized POI ligand (Figure 9). The precursors have a smaller molecular weight and thus an improved cell permeability. In the cytosol, CLIPTACs rapidly undergo an inverse electron-demand Diels-Alder cycloaddition (IEDDA) without the need for a catalyst to form the functional PROTAC molecule. The cells are incubated sequentially with the precursor molecules to avoid assembly reactions before entering the cells.

By means of this technique, BRD4 and ERK1/2 have been successfully degraded. Amazingly, no POI degradation occurred when the precursor molecules were reacted extracellularly to form the resultant CLIPTAC before cell incubation, indicating that the preformed CLIPTAC is not able to cross the cell membrane. This project marks an important step forward to achieve translation of PROTACs into clinical use. CLIPTACs demonstrate that it is possible to overcome the undesirable physicochemical properties of PROTACs for use in vivo and development into drugs. However, two prodrugs, which must not be mixed extracellularly, pose new challenges for pharmaceutical formulation, particularly regarding continuous dosing.

#### 2.1.4. HaloPROTACs

HaloTag is a genetically engineered protein tag derived from a bacterial dehalogenase and designed to react covalently with synthetic ligands bearing a chloroalkane [37]. HaloTag fusion proteins can be recombinantly expressed and have been widely used to label POIs in a bio-orthogonal fashion [37]. Since halogen alkanes only occur in the bacterial world, this is a selective targeting strategy in eukaryotes. By adding a reactive chloroalkane linker, several ligands have been synthesized for HaloTag fusion proteins to label them with a fluorophore or to attach an affinity tag [37], but they can also be used to recruit E3 ubiquitin-ligases. HaloPROTACs are PROTACs that induce the degradation of HaloTag fusion proteins. They generally comprise an E3 ligase ligand to which an alkyl chloride, typically hexyl chloride, is attached via a linker moiety (Figure 10a). Crews and co-workers have reported bifunctional HaloPROTACs which both bind the E3 ligase VHL and a HaloTag7 fusion POI (Figure 10b) [38]. HaloTag7 is a further development that has optimized the fusion proteins’ stability [39]. They used VL285 as a VHL ligand and installed a hexyl chloride separated by a PEG linker to form HaloPROTAC **13**, which robustly caused degradation of HaloTag7 hybrid proteins with GFP, ERK1, or MEK1 at nanomolar concentration [38].

Ishikawa et al. applied the concept of HaloPROTACs to cIAP1 and induced the degradation of the nuclear proteins CREB1 and c-Jun, which were deemed orphans [40,41]. Instead of using overexpression models, Ciulli, Alessi and co-workers used CRISPR/Cas9 genome editing technology to give rapid access to reversible degradation of HaloTag fusion proteins [42]. Furthermore, they enhanced the half-maximal degradation concentration (DC_50_) of **13** by subtle changes in the isoindolinone group, which was replaced by an *N*-acylamide of L-*tert*-leucine (**14**).

The advantages of HaloPROTACs are their modularity and the specific degradation of HaloTag7 fusion proteins, whereas native proteins are spared. The same PROTAC can be used to induce chemical knockdown of a range of proteins, provided that stable fusion proteins are available. However, HaloPROTACs only target genetically modified proteins, limiting clinical application. Additionally, since the hexyl chloride forms a covalent bond with the POI, degradation is not catalytic and requires stoichiometric amounts of the HaloPROTAC. Yet, HaloPROTACs are valuable tools for chemical genetics and allow to study the biological role of undruggable targets.

#### 2.1.5. PhosphoPROTACs

PROTACs constitutively trigger the POI’s ubiquitination and subsequent degradation regardless of cellular status. Phospho-dependent PROTACs (phosphoPROTACs) function as conditional mediators of protein knockdown and highly depend on cellular phosphorylation status (Figure 11) [43]. By means of phosphoPROTACs, one can precisely interfere with kinase signaling pathways and specifically target degradation in cells with activated pathways. The core component of phosphoPROTAC **15** is a short peptide sequence including a tyrosine, that can be phosphorylated by a specific receptor tyrosine kinase (RTK). After activation, the phosphorylated product can subsequently be recognized by a particular RTK downstream protein containing either a Src homology 2 (SH2) or a phosphotyrosine-binding (PTB) domain. PhosphoPROTACs have another peptide sequence, which is connected via an aminohexanoic acid linker and binds to VHL, as well as a poly-*D*-arginine tag to enhance cell permeability. The phosphoPROTAC only becomes a functional degrader when the signal pathway is activated, i.e., following phosphorylation by RTK. In such a manner, successful degradation was exemplified for fibroblast growth factor receptor substrate 2α (FRS2α) and phosphatidylinositol-3-kinase (PI3K).

As dysregulated RTK signaling can cause uncontrolled cell proliferation, phosphoPROTACs embody a tissue-selective strategy for cancer therapy and offer the ability to distinguish healthy cells from affected ones. However, these phosphorylation status-sensing PROTACs are compromised by low potencies in the two-digit micromolar range and poor pharmacokinetic properties such as low bioavailability after oral administration or half-life. PhosphoPROTACs may only exhibit biological or therapeutic effects in vivo, if protein degradation is a catalytic process and it must occur faster than the metabolism of the PROTAC. These effects will then last until the protein is resynthesized, even if the PROTAC has already been metabolized and excreted [43]. Due to their peptide character, it is currently unlikely that phosphoPROTACs will enter clinical use.

### 2.2. Hydrophobic Tagging (HyT)

Under physiological conditions, proteins fold to bury hydrophobic amino acid residues inside the protein and to expose hydrophilic moieties on the surface, which form stabilizing interactions with surrounding water molecules. The cellular quality control, i.e., chaperones, recognizes uncovered hydrophobic groups as a marker of protein misfolding and labels the protein for degradation by the ubiquitin-proteasome system (UPS). This unfolded protein response can be hijacked by hydrophobic tags (HyTs), which consist of a POI ligand and a hydrophobic fragment at the other end of the molecule connected by a linker. There are two plausible mechanisms, through which HyTs putatively induce proteasomal breakdown of their target proteins (Figure 12). HyTs could either be a direct motif recognized by chaperones and a hallmark of protein unfolding or the bulky and hydrophobic nature of the HyT truly leads to local or global changes in folding, which recruits chaperones as well.
Figure 12Ligands containing a hydrophobic tag (HyT) such as an adamantyl group bind the POI and induce its degradation through two putative mechanisms. Either HyTs mimic protein unfolding and recruit chaperones, which are responsible for protein quality assurance, and/or HyTs induce protein misfolding due to their bulky and hydrophobic nature. As a result in both cases, the POI is subjected to proteasomal degradation. Figure modified after Raina and Crews [46].
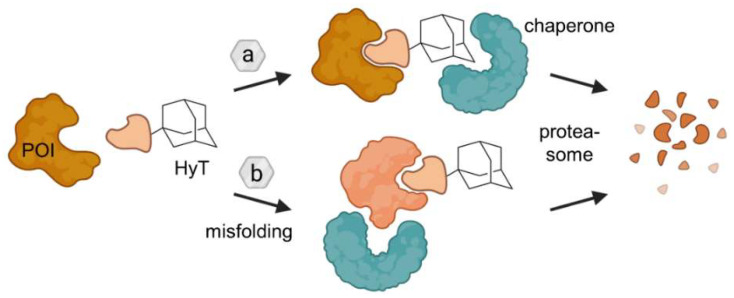


Initially, HyT technology has been established using HaloTag fusion proteins (*vide supra*) and used large and lipophilic tags such as an adamantyl group coupled to hexyl chloride (**16**, **17**) to selectively bind and fragment the fusion protein (Figure 13) [44,45]. The biggest shortcomings of this method are the need for genetically modified hybrid proteins and the covalent bonding, requiring stoichiometric amounts of HyT ligand rather than catalytic ones.

Hedstrom and colleagues serendipitously discovered triple *tert*-butyl carbamate-protected arginine (Boc_3_Arg) as a HyT group capable of degrading endogenous proteins [47]. They showed that if the degron is attached to the covalent ligand ethacrynic acid (**18**) or non-covalent inhibitor trimethoprim (**19**), glutathione-*S*-transferase (GST) and dihydrofolate reductase, respectively, are efficiently degraded. Their results indicated that the degradation was mediated by the proteasome but did not depend on ATP nor ubiquitin. The purified core 20S proteasome was sufficient to achieve POI degradation in a Boc_3_Arg-dependent manner and likely, the POI is pulled into the barrel-shaped proteasome by direct interactions between the Boc_3_Arg and the 20S proteasome [48].

In the light of this work, the HyT technology using adamantyl groups was also extended to degrade naturally occurring proteins such as human epidermal growth factor receptor 3 (Her3, also known as ErbB3), which was believed to be undruggable, as the HER3 gene codes for a catalytically inactive pseudokinase [49,50]. The adamantane moiety of **17** was coupled to a covalent ligand of the pseudokinase to yield partial Her3 degrader TX2-121-1 (**20**). Moreover, **20** inhibited the heterodimerization of Her3 with Her2 and c-Met, thus suppressing Her3-dependent signal and growth, which was not observed with the parent ligand [49]. Although covalent HyTs still depend on stoichiometric amounts, this intriguing progress inspired researchers to apply the concept to different targets [51,52].

Originally developed as a selective estrogen receptor modulator (SERM), the FDA-approved breast cancer drug fulvestrant (**21**) was found to degrade estrogen receptor α (ERα) by causing conformational changes which expose hydrophobic side chains [53]. Therefore, misfolded ERα is subjected to proteasomal lysis. Fulvestrant (**21**) contains a vast alkyl side chain encompassing a sulfoxide and, driven by its therapeutic success, several other selective estrogen receptor downregulators (SERDs) have been identified [54]. Androgen receptors (ARs) are closely related nuclear hormone receptors and have been tackled to make novel therapies for androgen-related disorders. Crews and co-workers were the first to demonstrate the adamantane-based HyT method using a non-covalent binder of AR [55]. SARD279 (**22**) is a selective AR downregulator (SARD) derived from a potent AR agonist to which adamantane was coupled via a PEG linker, turning agonistic activity into an antagonistic degrader. Despite this significant progress to extend HyT technology to clinical application, there are caveats concerning supposedly low plasma concentrations because hydrophobic compounds are usually bound to serum proteins to a high degree. In addition, the physicochemical and pharmacokinetic properties of HyTs pose a challenge to drug discovery. By contrast, traditional PROTACs have a known and defined mechanism, and appear to be closer to clinical application.

### 2.3. Autophagy-Targeting Chimeras (AUTACs)

Besides UPS, the second major protein degradation pathway under physiological conditions is macroautophagy (hereafter called autophagy). While UPS eliminates short-lived proteins and soluble misfolded proteins, selective autophagy is responsible for the clearance of intracellular debris including long-lived proteins, insoluble misfolded proteins such as protein aggregates and even bulky cytoplasmic organelles (e.g., mitochondria, peroxisomes) [9,56]. There are several mechanisms how selective autophagy occurs unified by the principle of autophagy receptors recognizing the cargo (Figure 14a). The latter is engulfed by the double-membraned autophagosome, which subsequently fuses with a lysosome and the vesicle load undergoes proteolysis.

Like PROTACs hijacking the UPS system, Arimoto et al. harnessed autophagy to induce POI degradation using so-called autophagy-targeting chimeras (AUTACs) [57]. They identified *S*-guaninyl PTM of proteins as a tag for selective autophagy and thereupon designed *p*-fluorobenzylguanine-based chimeric molecules containing known ligands for methionine aminopeptidase 2 (MetAP2, **23**) or FKBP12 (**24**) coupled via a PEG linker-D-cysteine motif (Figure 14b). Incubation of HeLa cells with these AUTACs resulted in the K63-linked polyubiquitination of their target and consequent silencing by autophagic degradation. AUTACs are generally limited to cytosolic POIs, but excitingly, 2-phenylindole-equipped AUTAC **25** targeted whole cell organelles [57]. It binds the translocator protein (TSPO) on the outer mitochondrial membrane and ubiquitination-dependently induced degradation of fragmented mitochondria in fibroblasts derived from a trisomy 21 patient [57]. Mitochondrial homeostasis could be restored by promoting mitochondrial biogenesis. These initial results are promising toward a potential clinical application. However, AUTAC mode of action needs to be further studied to gain more insight into limitations such as the likely demand of stoichiometric amounts of the AUTAC.

Similar to AUTACs, autophagy-tethering compounds (ATTECs) can target biomolecules for autophagosomal degradation [58]. They are based on ligands for the microtubule-associated proteins 1A/1B light chain 3B (LC3), which is a key player in autophagy. For the first series of ATTECs, the LC3 ligands GW5074 (in **26**) or 5,7-dihydroxy-4-phenylcoumarin (**27**) were used to develop heterobifunctional compounds with C_10_ alkyl linkers to tether oil red O-derived compounds (Figure 14c). These ATTECs bind lipid droplets and subject them to degradation via autophagy. ATTECs **26** and **27** cleared lipid accumulation in a cellular system and mouse models of obesity and non-alcoholic steatohepatitis (NASH). Moreover, liver fibrosis, another hallmark of NASH, was alleviated in the animal model, indicating promising effects for metabolic disorders. Unlike AUTACs, ATTEC-mediated degradation is not dependent on K63 ubiquitination and therefore not limited to protein targets [58]. Apart from lipid droplets, the concept can be applied to protein and non-protein substrates such as nucleic acids or organelles [58]. It will be interesting to see what impact the linker length has on ATTEC function and whether ATTECs can be developed for pharmacotherapy.

## 3. Deubiquitination

The ability to hijack the UPS for the proteasomal degradation of target proteins has been shown by PROTACs and has opened new opportunities to temporarily eliminate proteins. This can be exploited to either gain more insight into ‘undruggable’ protein’s function or to achieve a therapeutic effect by enzyme inhibition. However, a variety of proteins such as the tumor suppressor p53 are degraded rapidly under physiological circumstances due to ubiquitination [59]. Deubiquitination of such substrates might prove as alternative for cancer therapy and was the subject of a joint research project between the University of California, Berkeley and Novartis [60]. Following the example of PROTACs, the aim of the project was to develop deubiquitinase-targeting chimeras (DUBTACs). Deubiquitinases (DUBs) are a class of proteins which cleave ubiquitin from their substrate and thus are the physiological counterpart to the ubiquitination system. The simultaneous recruitment of DUBs and a POI by heterobifunctional molecules could induce the proximity required for deubiquitination of the target protein, prevent degradation and prolong their half-life. For the proof-of-principle, Henning et al. took a chemoproteomic approach to identify a ligand for the K48 ubiquitin-specific deubiquitinase OTUB1 and succeeded in finding a ligand which targets cysteine C23 covalently (Figure 15) [60]. By binding this allosteric site, the DUB activity of OTUB1 was not inhibited. Based on the covalent OTUB1 ligand, they then designed DUBTAC **28** using lumacaftor, a marketed corrector of the cystic fibrosis transmembrane conductance regulator (CFTR). Deletion of phenylalanine at position 508 leading to ΔF508-CFTR is the most common mutation in cystic fibrosis and destabilizes the protein, which gets ubiquitinated and degraded. DUBTAC **28** however, prevents CFTR from being degraded and leads to an accumulation of protein. Furthermore, **28** improved CFTR-dependent transepithelial conductance in primary cells from a cystic fibrosis patient bearing the ΔF508-CFTR mutation, suggesting that levels of functional CFTR were enhanced at the cell surface, too. To corroborate the validity of target protein stabilization with DUBTACs, the authors developed another class of DUBTACs for the tumor suppressor kinase WEE1, of which **29a** and **29b** increased WEE1 protein concentrations in a hepatoma cancer cell line to similar levels as after treatment with proteasome inhibitor bortezomib.

DUBTACs can be conceptualized to study ubiquitin-dependent cellular processes as well as to accumulate target proteins like tumor suppressors including p53 to counteract cancer cell proliferation. To corroborate the therapeutic potential of DUBTACs, Jin, Wei, and co-workers established a platform of DUBTACs for different transcription factors (p53, FOXO3A, and IRF3) which function as tumor suppressors [61]. These DUBTACs contain oligonucleotide sequences based on the corresponding DNA-binding motif to interact with the transcription factor. However, due to the relatively low specificity of the DNA-binding motifs, not only the target tumor suppressors but also non-specific proteins were stabilized. Furthermore, the DNA oligonucleotide moiety of these DUBTACs could inhibit target gene transcription and restrict tumor suppression. Hence, future research should focus on allosteric small molecular transcription factor ligands. It is doubtless that DUBTACs enrich the chemical toolbox of small molecule PTM inducers and they hold a promising potential to provide therapeutic benefit in diseases with destabilized or actively degraded proteins.

## 4. Phosphorylation and Dephosphorylation

Although under physiological conditions, ubiquitination is not the most frequent PTM, it has been the focus of pioneering work to alter the PTM profile of target proteins using heterobifunctional molecules. The success story of PROTACs gives rise to the question whether heterobifunctional small molecules could be engineered to modulate other PTMs. Among the many types identified so far, phosphorylation is the most frequent PTM and involves serine, threonine, and tyrosine residues [4,62]. The phosphate group introduces a negative charge and enhances the hydrophilicity of POIs, altering the interaction with surrounding molecules. Although a very small moiety, phosphate groups can have vast conformational consequences and regulate protein function [62]. While kinases transfer phosphate groups onto their substrates, the key enzymes of dephosphorylation are called phosphatases. The interplay of kinase and phosphatase activity regulates many cellular pathways and its disturbance is linked to diseases such as various types of cancer [62].

### 4.1. Phosphatase-Recruiting Chimeras (PhoRCs)

The crucial step of PROTAC-mediated POI degradation is the formation of a ternary complex, bringing POI and E3 ligase into close proximity. Generally, the E3 ligase could be replaced by any effector protein to posttranslationally modify POIs. Staben and colleagues aimed to design bifunctional molecules to enable POI dephosphorylation based on this proximity hypothesis (Figure 16a) [62]. As POI, the oncogenic kinases AKT (protein kinase B, PKB) and EGFR were selected because their kinase activity is regulated by phosphorylation state. The dephosphorylation of these kinases inhibits the signaling cascade by stopping both reception and transmission of the phosphate signal. As a proof-of-concept, first a covalent EGFR inhibitor (**30**) and an allosteric AKT inhibitor (**31**) each were coupled to a PEG linker and a terminal hexyl chloride (Figure 17). Both **30** and **31** reduced phosphorylated POI (pPOI) levels in cell lines overexpressing a HaloTag fusion protein of protein-phosphatase 1 (PP1) [62].

To study target dephosphorylation in a less artificial system, they then prepared phosphatase-recruiting chimeras (PhoRCs). PhoRCs are heterobifunctional molecules, which form a ternary complex with their pPOI and a phosphatase (Figure 16a). Subsequently, the phosphatase detaches the phosphate group from the POI and, given that the PhoRC does not bind the proteins irreversibly, the ternary complex dissociates. The PhoRC is released and can undergo further catalytic cycles [62]. The group of researchers around Staben developed PhoRCs based on the AKT inhibitor present in **31** and a PP1-activating peptide fused by a PEG linker [62]. The desired effects were achieved using minimum PP1 recognition motif RVSF in PhoRC **32**, which reduced levels of phosphorylated AKT at both T308 and S473 in LNCaP cells. When using an ATP-competitive AKT inhibitor for the PhoRC (**33**), rather than the allosteric AKT inhibitor, the effect was less pronounced. That is not surprising because ATP-competitive inhibitors are known to increase pAKT^T308^ and pAKT^S473^, thus counteracting the dephosphorylation induced by PhoRC **33** [62].

These results highlight phosphorylation as PTM that can be modulated by chimeric molecules in vitro. However, there remain obstacles to clinical application. Incorporation of the RVSF motif may compromise interaction of the ubiquitous PP1 with interacting proteins. Furthermore, the peptide moiety in the PhoRC does not obey Lipinski’s Rule of Five (low logP, high number of hydrogen bond donors and acceptors) [19]. Besides having a poor oral bioavailability, peptides are also prone to proteolytic cleavage and hence have short half-lives [11]. This might explain why the PhoRCs were only active at relatively high concentrations. Unlike phosphoPROTACs (*vide supra*), which become active in the presence of phosphorylation, PhoRCs likely bind their target protein regardless of its phosphorylation status. The fraction of PhoRC bound to unmodified, non-phosphorylated POI reduces the effective PhoRC concentration in its working environment. Therefore, a strategy to specifically address pPOI would create more potent dephosphorylating agents. Additionally, a small molecule allosteric phosphatase ligand could overcome pharmacokinetic problems at least partially and drive clinical development.

### 4.2. Phosphorylation-Inducing Chimeras (PhICs)

Recently, the counterpart of PhoRCs were disclosed by the Choudhary lab and termed phosphorylation-inducing chimeric small molecules (PhICs) [63]. Based on a similar mechanism as PROTACs, these bifunctional compounds complement the toolbox to modulate POI phosphorylation status by hijacking kinases (Figure 16b). PhICs bring their target protein and a specific kinase close together so that the latter can catalyze the transfer of a phosphate group from ATP onto the target protein, thereby releasing ADP as a side product. Unless the PhIC comprises covalent ligands of the respective kinase and target proteins, it can act in a catalytic fashion and install multiple copies of phosphate on the POI. The Choudhary lab first realized this concept by designing PhIC **34** featuring the well-known BET bromodomain ligand (+)-JQ1 and a synthetically more amenable derivative of an allosteric AMP-activated protein kinase (AMPK) activator (Figure 18). Both ligands are tied together by a click-chemistry-based triazole linker. PhIC **34** induced both native phosphorylation at the natural site Ser484/488 and neophosphorylation at various sites of BRD4, which is not a natural substrate of AMPK. In an analogous manner to (+)-JQ1-containing PROTACs [26,32], PhICs can achieve selectivity among paralogues and subtypes which are not observed for their parent POI ligand. For instance, **34** selectively mediated the phosphorylation of BRD4 over BRD2 and BRD3 [63].

To prove the wide applicability of this approach, the authors designed a second PhIC (**35**) based on (+)-JQ1 and a protein kinase C (PKC) activator, which phosphorylated BRD4 to a high degree in cooperation with PKCα, but only moderately with isoforms PKCβI and II and to a minor extent with PKCγ and PKCδ. Furthermore, phosphorylation occurred on BRD2 and BRD4, but not on BRD3.

Although PhICs **34** and **35** demonstrated promising initial results, phosphorylation was not observed in cells and the authors assume the different subcellular localization of BRD4 (nucleus) and kinases (cytosol) to play a factor. Hence, a third PhIC (**36**) was synthesized employing the cytoplasmic BTK as POI and AMPK as a kinase, which naturally do not interact with each other. In the presence of **36**, HEK293T cells transfected with BTK showed increased levels of pBTK^S180^, thus rendering PhICs as a valuable approach to induce PTMs in cellulo [63].

PhoRCs and PhICs are complementary bifunctional molecules that allow the precise control of their target protein’s phosphorylation status and may soon be applied to novel concepts to tackle kinase-dependent cancer pathways or to weaken the DNA interaction of transcription factors by charging them negatively. PhICs certainly expand the toolbox to adjust protein function by tailoring their PTM profile and will hopefully be translated into clinical application for novel therapies.

## 5. Acetylation

Lysine acetylation is a key PTM which, like methylation, eliminates the positive charge on the ε-amino group and has significant impact on the protein’s electrostatic properties [64]. Acetylation is a covalent modification and reversible. A protein’s acetylation state is regulated by histone or lysine acetyltransferases (HATs/KATs) and histone/lysine deacetylases (HDACs/KDACs). There are functional consequences from acetylation that can be illustrated using histone proteins. Lysine residues are protonated under physiological conditions endowing a positive charge on the protein surface to enable electrostatic forces between the histone and the negatively charged phosphate backbone of DNA. Consequently, binding of DNA to histones is tight. Upon acetylation, the positive charge on the histone surface is eliminated and the binding to DNA is loosened. Acetylation can be additionally sensed by bromodomains and other proteins.

Given the fundamental role of acetylation in cellular processes, Wang et al. wanted to develop a chemical tool to induce acetylation of target proteins [65]. In a proof-of-concept study, they expressed POI fusion proteins with the genetically engineered FKBP12 variant FKBP12^F36V^ originating from a bump-and-hole approach [66]. This variant can be specifically targeted with a bumped ligand. For their so-called acetylation-tagging system (AceTAG), the authors synthesized a heterobifunctional molecule based on the bumped ligand and connected it via a PEG linker to a ligand for the KATs CREBBP and p300 (Figure 19) [65]. CREBBP and p300 are close paralogues and together regulate approximately two thirds of acetylation sites in human, so this approach might provide a modular acetylation method. Like for other heterobifunctional molecules, the hypothesis was pursued that by inducing proximity to a KAT, a target protein can be acetylated. After initial experiments investigating the linker length and composition, AceTAG-1 (**37**) proved to be the most efficient acetylation-inducing compound. Using **37**, the corresponding fusion proteins of histone H3.3, the NF-κB subunit p65/RelA, and the tumor suppressor p53 were acetylated. Astonishingly, acetylation was detected after only five minutes and went down to initial levels two hours after washout of **37**, probably due to opposing HDAC activity. Moreover, H3.3 acetylation was selective and preferentially occurred on K18, K23, and K27 [65]. However, there are limitations to the AceTAG system, namely the requirement of a KAT ligand that does not block the catalytic site. CREBBP and p300 are ideal for this purpose because they comprise multiple domains of which the bromodomain is targeted by AceTAG-1 (**37**). Another limitation is the availability of antibodies to study the acetylation site. The AceTAG system not only has big potential as a tool for chemical biology but might be also used as a therapeutic strategy. There is a long way before this can be achieved and the next step must be to develop this concept further for application to endogenous proteins.

## 6. Excursus: Induced Degradation of Extracellular and Surface Proteins and RNA

In addition to the already treated small molecular chimeras that induce PTMs, there is other TACnology that is either no longer small molecules (e.g., antibody-based LYTACs) or does not target proteins such as RIBOTACs. These molecules shall be covered in the following excursus.

### 6.1. Lysosome-Targeting Chimeras (LYTACs) and Endosome-Targeting Chimeras (ENDTACs)

Although PROTACs have a broad potential substrate scope, they are limited to intracellular target proteins. This strategy remains elusive for extracellular proteins and their membrane-bound cognate receptors, which constitute important physiological signaling pathways and are not affected by UPS [67]. Those potential targets including cytokines, chemokines, and growth factors bind receptors at the cell surface and can be inhibited by monoclonal antibodies, whereas small molecules have been less successful [68].

Lysosome-targeting chimeras (LYTACs) allow the knockdown of extracellular and membrane-bound proteins such as apolipoprotein E4, EGFR, CD71 and programmed death-ligand 1 [69]. In contrast to ENDTACs (*vide infra*), they are based on antibodies. LYTACs comprise a multivalent glycopolypeptide agonist for the cation-independent mannose-6-phosphate receptor (CI-M6PR, also known as insulin-like growth factor 2 receptor (IGF2R)) coupled to an antibody against the secreted or membrane POI (Figure 20a,b). Upon binding to the LYTAC, CI-M6PR internalizes and shuttles the complex to the lysosome, where the cargo is degraded, while the receptor itself is recycled and heads back to the cell surface [69]. This method is generally applicable to shuttling receptors other than CI-M6PR. A key challenge for translation into clinical application will be to improve pharmacokinetic parameters of LYTACs [69].

Crews et al. hypothesized that secreted proteins can be internalized and degraded via the endolysosomal pathway using heterobifunctional small molecules termed ‘endosome-targeting chimeras’ (ENDTACs) [68]. Such a double-faced compound binds the extracellular POI as well as a decoy receptor that is hijacked to internalize the ternary complex by endocytosis (Figure 20c). Subsequently, the POI buds off the cell surface within the early endosome, which matures to the late endosome, where the decoy surface receptor releases its cargo. Ideally, the membrane-bound receptor would then be able to recycle to the cell surface and close the catalytic cycle. Eventually, the target protein is degraded in the lysosome. In contrast to LYTACs, the POIs are targeted by small molecules rather than antibodies. Unfortunately, the publication disclosing the first ENDTACs had to be retracted [68]. Nevertheless, the concept of the study is a valid approach and ENDTACs will hopefully soon bridge the gap for induced extracellular protein degradation. ENDTACs embody the potential to deplete disease-related secreted proteins.

### 6.2. Ribonuclease-Targeting Chimeras (RIBOTACs)

So far, different TACnologies have mainly focused on PTMs of proteins as a target. The Disney lab added RNA degraders to the toolbox of heterobifunctional small molecules [70]. The most common way to target RNA is to use antisense oligonucleotides for unstructured RNA regions. As a consequence, ribonuclease (RNAse) H is recruited, and the RNA target is degraded [70].

Researchers in the Disney lab developed small molecule RNA binders that overcome the poor bioavailability and biostability of antisense oligonucleotides [70]. Their strategy is to target precursor microRNA hairpins (pre-miR), which feature regions with defined structures accessible to small molecules. Pre-miR is cleaved in the cytoplasm to form mature microRNA (miR). Ribonuclease-targeting chimeras (RIBOTACs) are bifunctional molecules comprising a small molecule pre-miR binder and a short 2′-5′-linked polyadenylate unit that recruits latent ribonuclease (RNAse L). RIBOTACs bring their target RNA and RNAse L into close proximity, inducing the degradation of pre-miR (Figure 21a). Specifically, the first RIBOTAC (**38**) targets pre-miR-210, which is overexpressed in hypoxic tumors. The RNA-targeting moiety of **38** is the small molecule Targapremir-210 (Figure 21b) and coupled to a 2′-5′-linked tetraadenylate unit. Levels of both pre-miR-210 and miR-210 in hypoxic MDA-MB-231 cells decreased upon treatment with **38** and apoptosis was induced. While the parent compound Targapremir-210 only had a narrow 5-fold selectivity window over off-target DNA, no DNA binding was observed for RIBOTAC **38** in the presence of RNAse L. Moreover, **38** did not significantly affect off-target RNA [70]. Given the abundance and variety of RNA, these results are ground-breaking for the development of small molecule RNA-targeting drugs. However, the pharmacokinetic profile of the nuclease-recruiting moiety needs to be optimized for clinical application.

## 7. Conclusions and Outlook

In the 20 years since the PROTAC technology has first been reported [71], this field of research has rapidly grown. Peptidic molecules have turned into heterobifunctional compounds with oral availability and academic basic research has translated into industrial development. With the first PROTACs under clinical investigation (phase I and II), we are entering a critical phase for heterobifunctional molecules in general. PROTACs will be the pioneers for heterobifunctional molecules and the future of such compounds will depend on the success of PROTACs in the clinic. Although clinical PROTAC development is still in its infancy, interim results indicate that safety, tolerance, and exposure can be provided [72]. It has been shown for the first time, that PROTACs mediated the degradation of their target protein in humans [72]. However, our current understanding of pharmacology of heterobifunctional molecules is very limited and predictions tools for oral availability are scarce. We need to closely monitor the further clinical development of PROTACs and to apply these lessons to other TACnologies.

As demonstrated by RIBOTACs, recent advances in TACnology are not limited to proximity-induced PTMs, but incorporate the potential to lure a multitude of effector proteins to a protein, RNA, or other biomolecular targets and to trigger desired effects in a controlled and precise manner [70]. DNA sequence defines a protein’s amino acid sequence and consequently encodes the totality of protein properties including biological activity, cellular localization, and interactions to other proteins. Due to the preserved genetic code, responses to environmental and developmental influences can only be made quickly on protein level rather than on DNA level. PTMs are Nature’s method to fine-tune protein properties [1]. In many diseases, especially chronic disorders, the body cannot cope with pathological phenomena and requires external intervention [5]. Tailor-made PTMs reshape proteins in a defined way and could contribute to drug discovery. To date, small molecule-based heterobifunctional chimeras only address ubiquitination (PROTACs, AUTACs, DUBTACs), phosphorylation (PhoRCs/PhICs), and acetylation (AceTAG), leaving a wide gap for other common PTMs (Figure 1). Glycosylation is the second most frequent PTM and involved in protein folding, circulatory half-life, cellular homeostasis, and cell adhesion in the immune system [73]. To name but one possible application, deglycosylation-targeting chimeras might be used as viral entry inhibitors.

Methylation and demethylation of lysine and arginine are common PTMs, but not limited to histone proteins and affect protein charge as well as PPIs [74]. Heterobifunctional molecules that induce (de-)methylation would be particularly interesting in the field of epigenetics. The possibilities of small molecule-inducers of PTM seem to be almost endless and give access to innovative therapeutic concepts. The enhancement of aqueous solubility, cell-permeability, and half-life will be crucial in the future and would offer undreamt possibilities for medicinal chemistry, and clinical application in precision medicine.

## Figures and Tables

**Figure 1 molecules-28-00690-f001:**
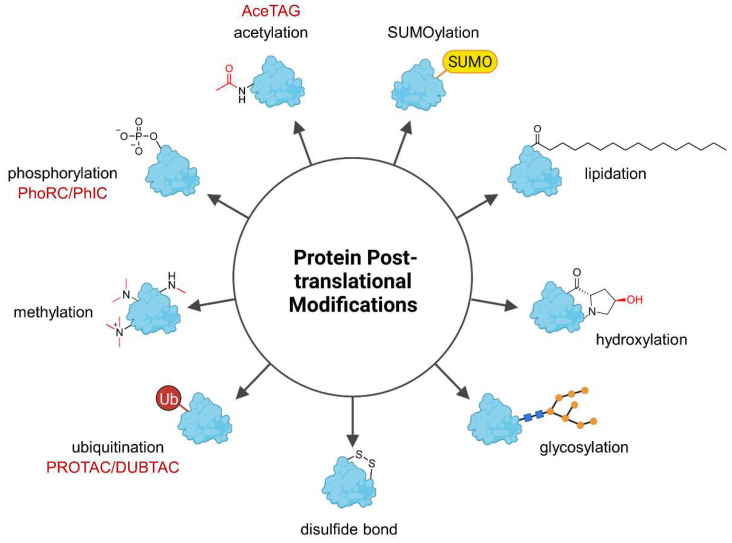
Overview of the most common posttranslational protein modifications (PTMs). PTMs have versatile impact on proteins and affect physicochemical properties such as hydrophilicity and overall charge as well as protein localization but they can also flag a protein for degradation (ubiquitination). Heterobifunctional molecules can precisely trigger the specific modification of their target protein, but to date only (de-)ubiquitination, (de-)phosphorylation, and acetylation have been modulated by the indicated chimeras (in red). AceTAG—acetylation-tagging system, DUBTAC—deubiquitinase-targeting chimera, PhIC—phosphorylation-inducing chimera, PhoRC—phosphatase-recruiting chimera, PROTAC—proteolysis-targeting chimera, SUMO—Small ubiquitin-related modifier, Ub—ubiquitin.

**Figure 2 molecules-28-00690-f002:**
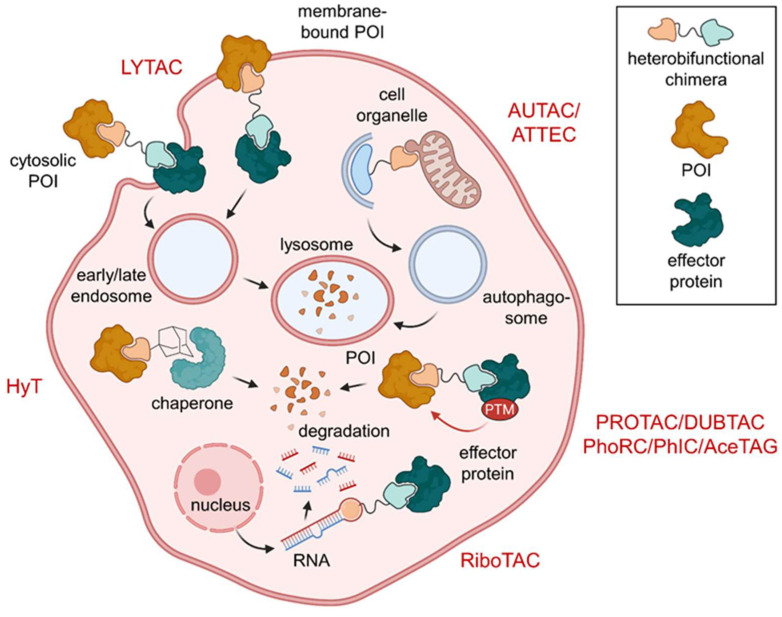
Overview of heterobifunctional molecules effecting PTMs. Lysosome-targeting chimeras (LYTACs) subject cytosolic or membrane-bound proteins of interest (POIs) to lysosomal degradation. Protein degradation can also be modulated by hydrophobic tags (HyT) and proteolysis-targeting chimeras (PROTACs). HyTs induce conformational changes that recruit chaperones and PROTACs effect degradation by polyubiquitination of their POI. On top of that, autophagy-targeting chimeras (AUTACs) and autophagy-tethering compounds (ATTECs) drag their POI or target organelle into the autophagosome for degradation. In addition to PROTACs, ribonuclease-targeting chimeras (RIBOTACs) mediate degradation of target hairpin RNA. Protein stabilization can be affected by deubiquitinase-targeting chimeras (DUBTACs). Phosphorylation status can be controlled using phosphorylation-inducing chimeras (PhICs) or phosphatase-recruiting chimeras (PhoRCs), while compounds based on the acetylation-tagging system (AceTAGs) can trigger POI acetylation.

**Figure 3 molecules-28-00690-f003:**
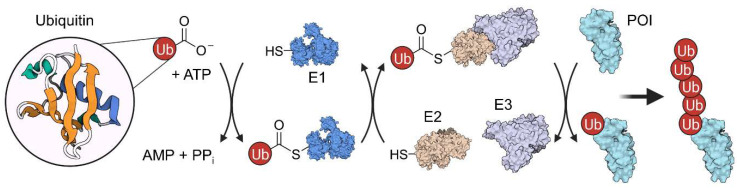
Ubiquitination mechanism involves three different enzymes (E1–E3) and is ATP-dependent. First, the ubiquitin peptide is transferred to an E1 enzyme under ATP consumption. Then, ubiquitin is further transferred to the E2 protein in an transthioesterification reaction. Subsequently, an E2:E3 complex forms, which can ubiquitinate the protein of interest (POI) in an iterative process.

**Figure 5 molecules-28-00690-f005:**
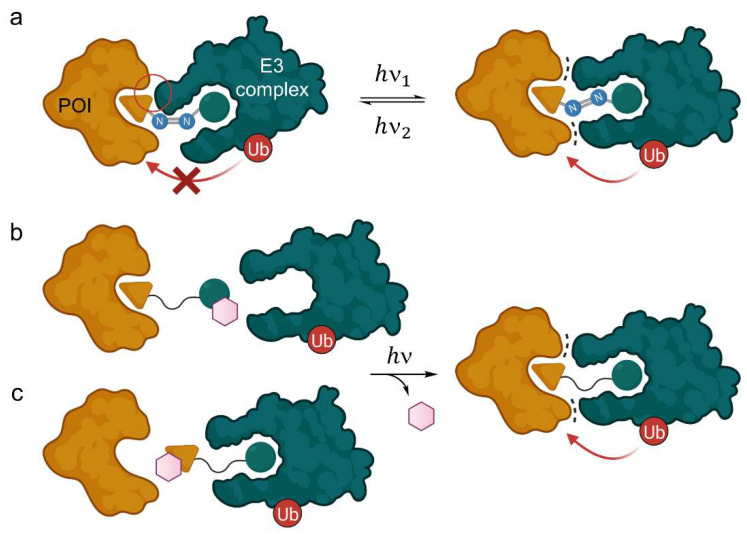
Light-responsive PROTACs and their mode of action. (**a**) Azo group-containing PROTACs can be irradiated to form either the *E*- or the *Z*-isomer, one of which is degrading and the other inactive, thus allowing spatiotemporal control of target protein knockdown. The distance between both ligands is significantly enhanced in the *trans*-configuration, where beneficial protein-protein interactions take place (dots), whereas the *cis*-PROTAC is too short to form the ternary complex, or *vice versa*. (**b**,**c**) PROTACs with an additional photocleavable group (purple hexagon) prevent either the E3 ligase (**b**) or the POI (**c**) from binding. Light irradiation cleaves the group irreversibly and releases the active PROTAC.

**Figure 6 molecules-28-00690-f006:**
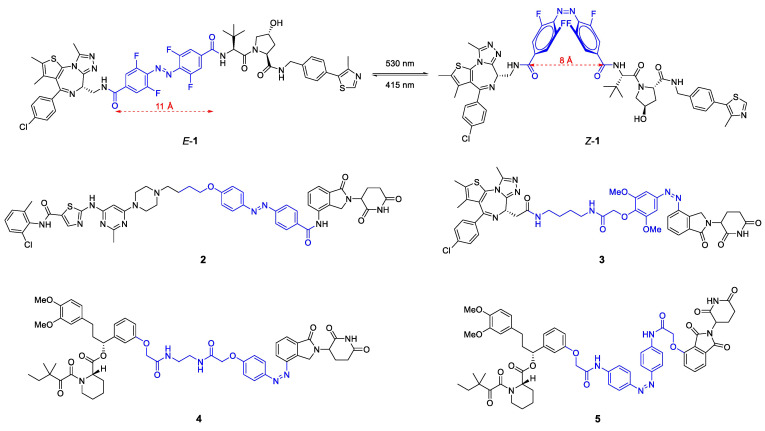
Chemical structure of photoswitchable PROTACs **1**–**5**. The linker containing the light-responsive azo group is highlighted in blue color. All molecules can be precisely converted to either the *E*- or *Z*-isomer of which only the active degrader is shown for **2**–**5**.

**Figure 7 molecules-28-00690-f007:**
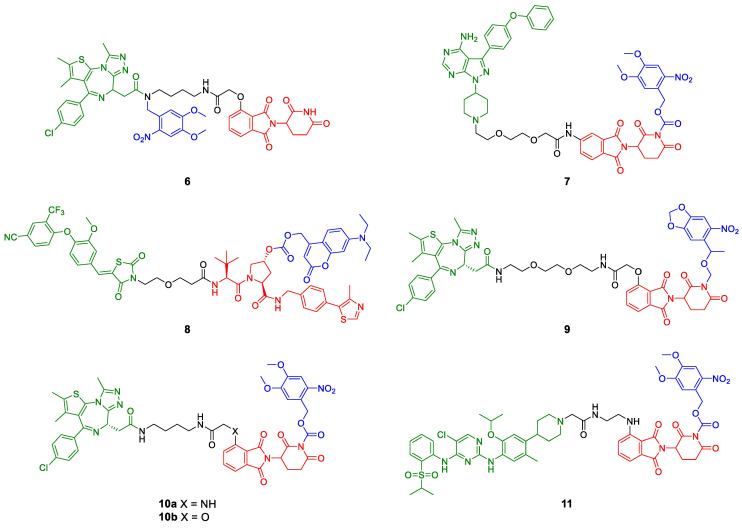
PROTACs with photocleavable moieties (blue). Bulky groups are added to functional PROTACs either at the POI ligand (green, **6**) or the E3 ligase ligand (red, **7**–**11**) and prevent or impair binding. Upon irradiation at a defined wavelength, the light-responsive moieties are cleaved, and the active degrader is released.

**Figure 8 molecules-28-00690-f008:**
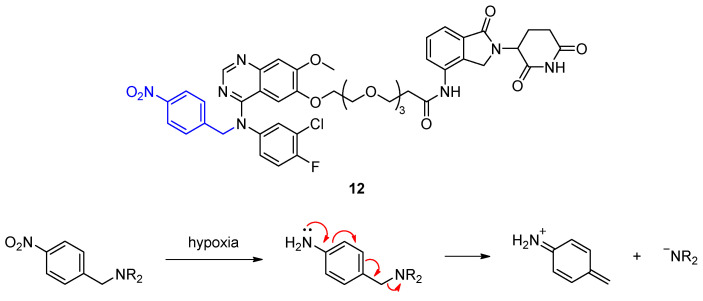
Hypoxia-activated EGFR PROTAC **12** selectively degrades its target under hypoxia, when the 4-nitrobenzyl group (blue) is reduced by nitroreductases and cleaved from the PROTAC to give the corresponding secondary amine.

**Figure 9 molecules-28-00690-f009:**
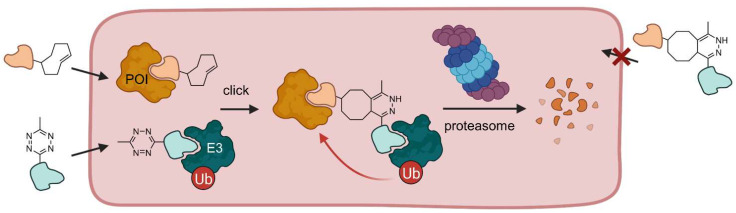
Click-formed PROTACs (CLIPTACs) can be generated in vivo. Cells are incubated with a cyclooctene-functionalized POI ligand and a tetrazine-derivatized E3 ligase ligand in a sequential manner. Both small molecules are cell permeable and undergo a click reaction in the cell. If the click reaction takes place beforehand, i.e., outside the cell, the PROTAC is not able to penetrate the cell membrane [36].

**Figure 10 molecules-28-00690-f010:**
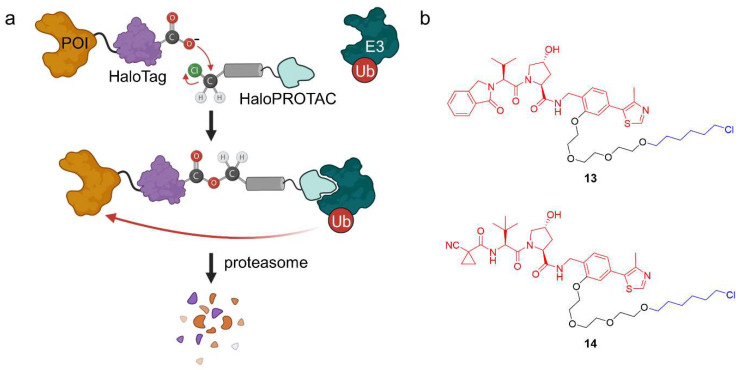
HaloPROTACs and their mode of action. (**a**) HaloPROTACs specifically target engineered HaloTag7 fusion POIs for degradation in a modular way. They bind to an E3 ubiquitin-ligase and, via the reactive hexyl chloride, to the HaloTag protein. (**b**) Chemical structures of two HaloPROTACs. The VHL ligand (VL285 in **13** and VH298 in **14**) is shown in red, the linker in black, and the hexyl chloride in blue.

**Figure 11 molecules-28-00690-f011:**
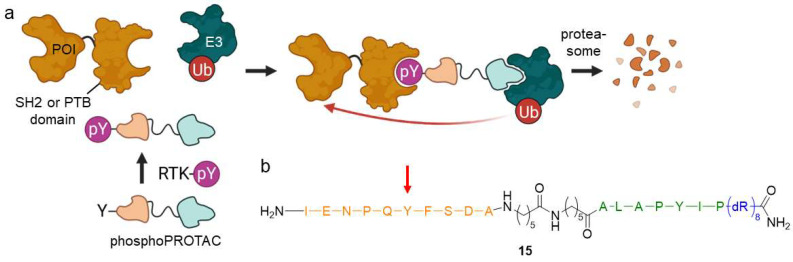
Phospho-dependent PROTACs (phosphoPROTACs) are controlled by the cellular phosphorylation state. (**a**) Upon phosphorylation by activated RTK, phosphoPROTACs bind to proteins containing a SH2 or PTB domain and an E3 ligase, marking the POI for proteasomal degradation. (**b**) PhosphoPROTACs are composed of a peptide sequence (orange), which can be phosphorylated at the tyrosine residue (red arrow) to bind a SH2 or PTB domain-containing protein, a spacer, a second peptide ligand for VHL (green), and a poly-*D*-arginine residue (blue) to enhance cell permeability.

**Figure 13 molecules-28-00690-f013:**
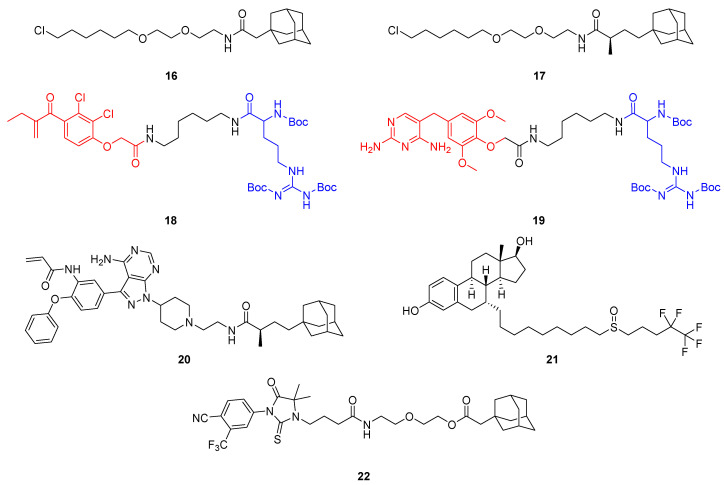
Hydrophobic tag (HyT)-containing bifunctional molecules that enable POI degradation via the proteasome. The HyTs in **16** and **17** are linked to hexyl chloride and bind HaloTag fusion proteins irreversibly. The Boc_3_Arg group (blue) can mediate POI degradation when bound to ligands (red) such as ethacrynic acid (**18**) or trimethoprim (**19**). Further examples of HyT-containing degraders are TX2-121-1 (**20**), the selective estrogen receptor downregulator (SERD) fulvestrant (**21**) and the selective androgen receptor degrader (SARD) SARD279 (**22**).

**Figure 14 molecules-28-00690-f014:**
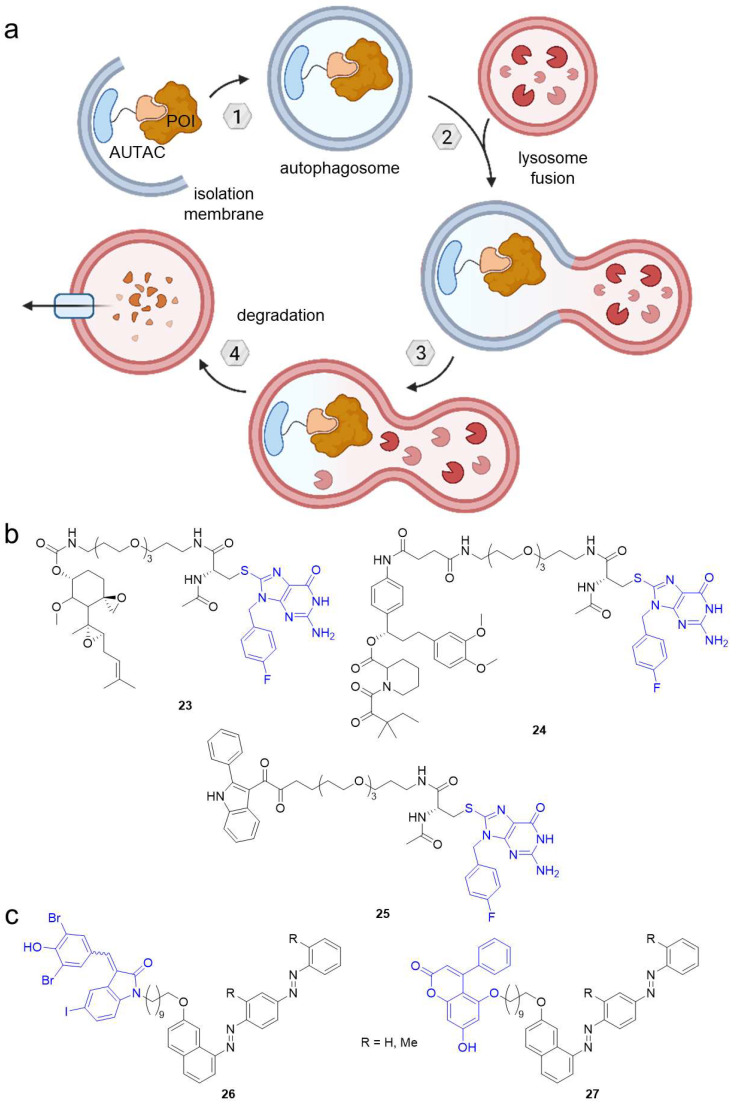
Autophagy-targeting chimeras (AUTACs) feature an *S*-guaninyl tag to target the POI for autophagosomal degradation involving K63-linked polyubiquitination. (**a**) AUTACs bind a POI and are recognized by autophagy receptors. The POI is wrapped by the autophagosome, which fuses with a lysosome. Eventually, the POI is degraded by proteolysis. (**b**) AUTACs targeting MetAP2 (**23**), FKBP12 (**24**), and TSPO (**25**). (**c**) Autophagy-tethering compounds (ATTECs) can target non-protein biomolecules like lipid droplets for autophagosomal degradation. The blue groups are ligands for autophagosome protein LC3. Using a C_10_ alkyl chain, the LC3 ligands GW5074 (**26**) and 5,7-dihydroxy-4-phenylcoumarin (**27**) were coupled to Oil Red O derivatives binding lipid droplets.

**Figure 15 molecules-28-00690-f015:**
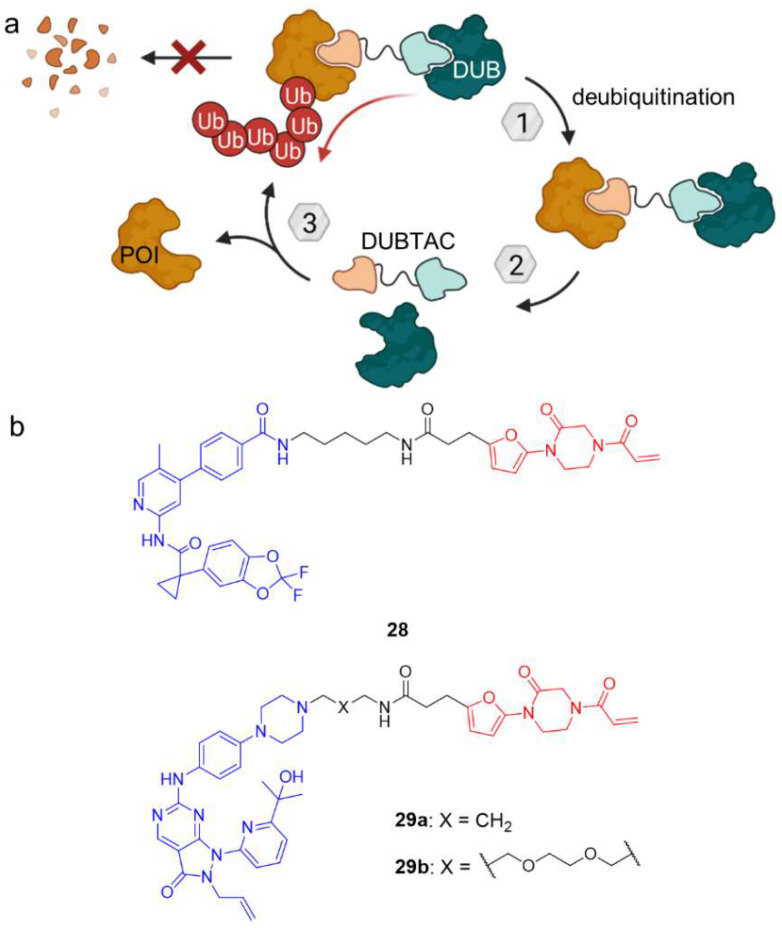
(**a**) Deubiquitinase-targeting chimeras (DUBTACs) mediate removal of ubiquitin degradation marks and thus stabilize target proteins. (**b**) DUBTACs **28** and **29** are composed of a POI ligand (blue), a linker unit, and a covalent allosteric ligand of the OTUB1 deubiquitinase (red). In case of **28**, the POI ligand is the corrector of the cystic fibrosis transmembrane conductance regulator (CFTR) lumacaftor, whereas **29** features an inhibitor of tumor suppressor kinase WEE1. In contrast to PROTACs, DUBTACs induce POI deubiquitination and thus stabilize protein levels.

**Figure 16 molecules-28-00690-f016:**
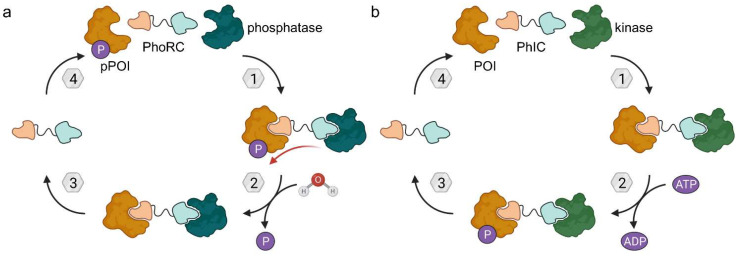
Phosphatase-recruiting chimeras (PhoRCs) and phosphorylation-inducing chimeras (PhICs). (**a**) PhoRCs are heterobifunctional molecules and bind the phosphorylated POI (pPOI) as well as a phosphatase. When the ternary complex forms, proximity enables the dephosphorylation of the POI. PhoRCs act catalytically unless the target proteins are bound irreversibly (covalently). (**b**) PhICs are the phosphorylating counterpart of PhoRCs and hijack kinases to phosphorylate their target consuming ATP.

**Figure 17 molecules-28-00690-f017:**
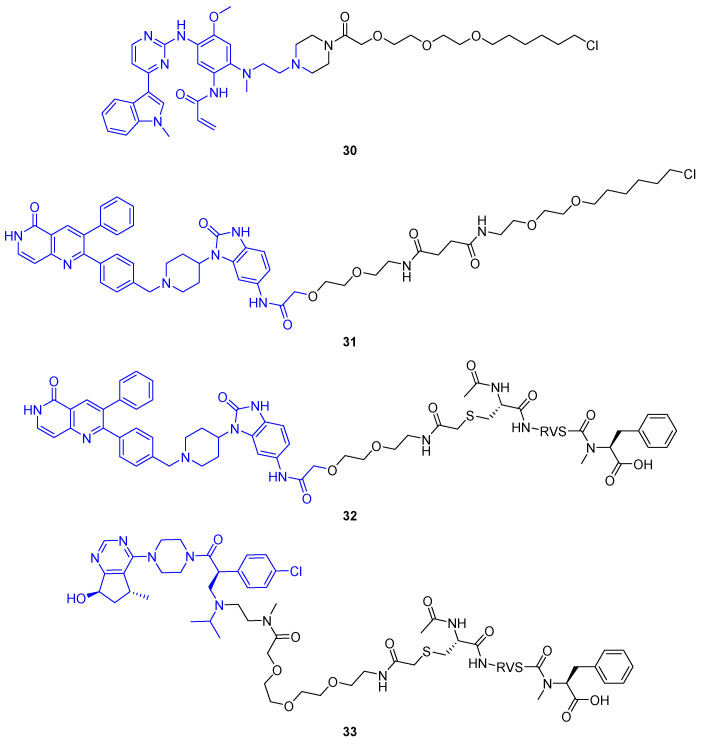
Dephosphorylation-targeting heterobifunctional molecules from Yamazoe et al. [62]. Initial compounds based on HaloTag-reactive groups induced dephosphorylation of EGFR (**30**) and AKT (**31**) fusion proteins. Additionally, PhoRCs **32** and **33** were developed to remove phosphate from pAKT in native cell lines. While **32** incorporates an allosteric AKT binder, **33** comprises an ATP-competitive ligand.

**Figure 18 molecules-28-00690-f018:**
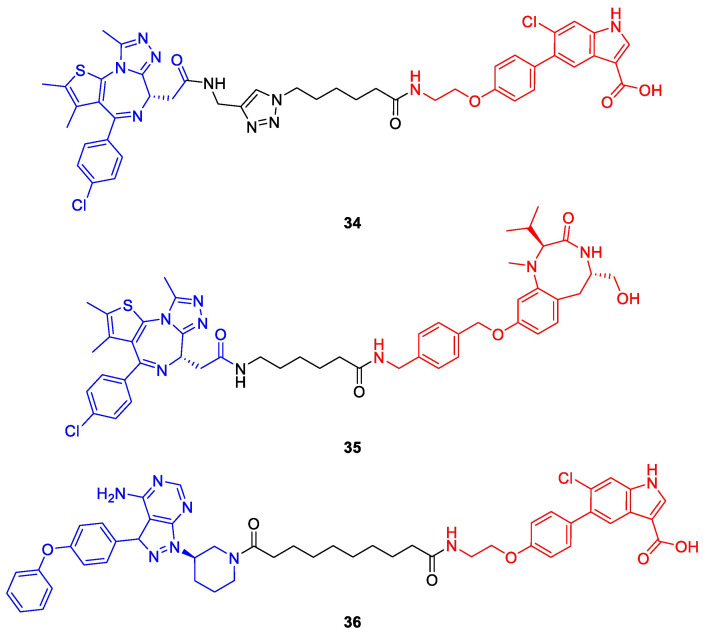
Phosphorylation-inducing chimeric small molecules (PhICs) reported by Siriwardena et al. [63]. Compounds **34** and **35** induce the phosphorylation of BRD4 by AMP-activated protein kinase (AMPK) or protein kinase C (PKC). In contrast to **34** and **35**, PhIC **36** is effective in living cells and facilitates BTK phosphorylation via AMPK. Kinase ligands are shown in red and target protein ligands in blue.

**Figure 19 molecules-28-00690-f019:**
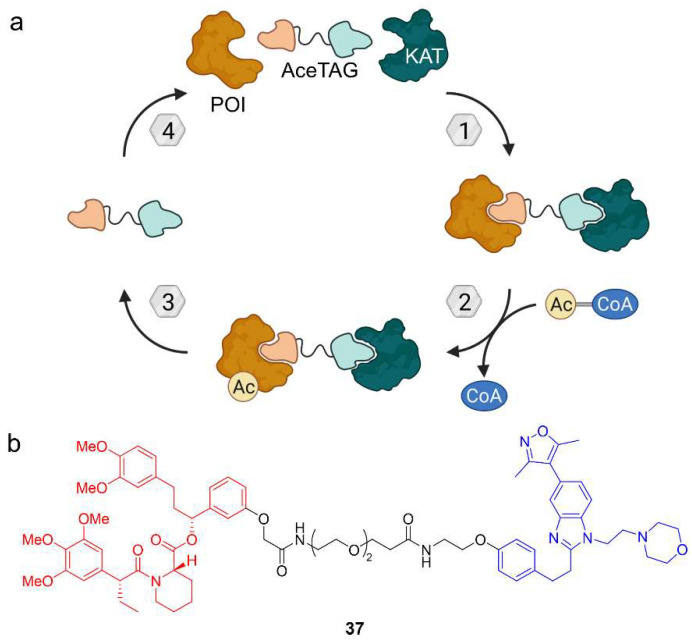
Target acetylation can be mediated by the acetylation-tagging (AceTAG) system. (**a**) Lysine acetyltransferases (KATs) are recruited to catalyze the transfer of an acetyl group from acetyl-CoA to the POI. (**b**) AceTAG-1 (**37**) is an acetylation-tagging molecule based on a FKBP12 ligand (red) and a ligand for the KATs CREBBP/p300 (blue). The CREBBP/p300 ligand binds the bromodomain and does not block the KAT domain for acetylation of FKBP12-based POI fusion proteins.

**Figure 20 molecules-28-00690-f020:**
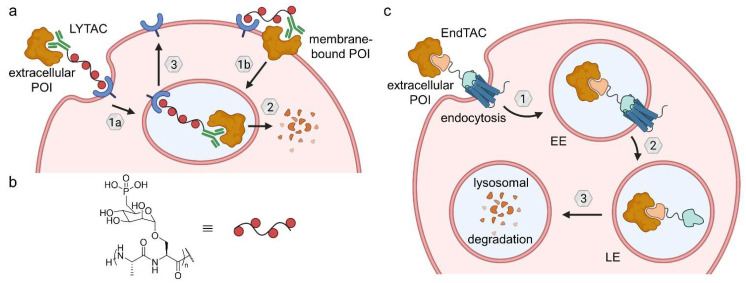
Lysosome-targeting chimeras (LYTACs) and endosome-targeting chimeras (ENDTACs). (**a**) LYTACs either target extracellular (1a) or membrane-bound POIs (1b), which are recognized by an antibody and trafficked to the lysosome (2) by a polyglycopeptide ligand for CI-M6PR. After lysosomal POI degradation, CI-M6PR returns to the cell surface (3). (**b**) The glycopeptide backbone is based on serine-*O*-mannose-6-phosphonate (M6Pn). (**c**) ENDTACs target secreted proteins and undergo endocytosis (1) mediated by a decoy GPCR, followed by lysosomal degradation (2 and 3). EE—early endosome, LE—late endosome.

**Figure 21 molecules-28-00690-f021:**
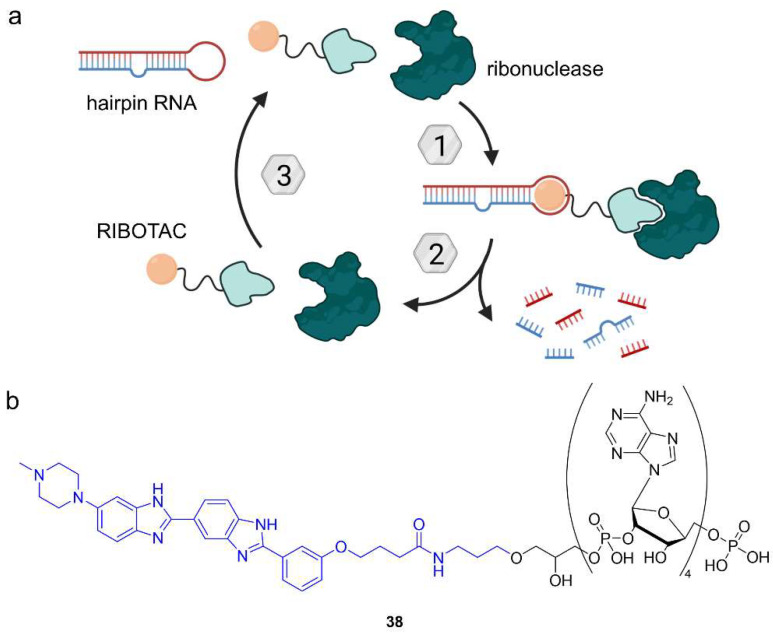
Ribonuclease-targeting chimeras (RIBOTACs) induce the ribonuclease-mediated degradation of RNA. (**a**) RIBOTACs comprise a small molecule binder of RNA and a 2′-5′-linked oligoadenylate tail as a recognition motif for ribonuclease. Formation of a ternary complex facilitates the proximity of RNA and ribonuclease, leading to RNA degradation. (**b**) Chemical structure of RIBOTAC **38** targeting pre-miR-210 using Targapremir-210 (blue).

## Data Availability

Not applicable.

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
