# Peer review of "Emerging TACnology: Heterobifunctional Small Molecule Inducers of Targeted Posttranslational Protein Modifications"

_molecules, 2023, doi:10.3390/molecules28020690_

Round 1

Reviewer 1 Report

Pascal Heitel addresses recent advances in the development of het- 11 erobifunctional small molecules that enable targeting or hijacking PTMs.The overall study is interesting, but can be potentially improved and several comments are listed below for consideration.

1. The theoretical basis of the article is ample, but research on the application of therapeutic drugs can be further supplemented.

2. The research on whether there is a negative side to the normal function of Heterofunctional Small Molecule In - 2 conductors should be supplemented in the original text.

Author Response

Response to Reviewer 1 Comments

Pascal Heitel addresses recent advances in the development of heterobifunctional small molecules that enable targeting or hijacking PTMs. The overall study is interesting, but can be potentially improved and several comments are listed below for consideration.

Point 1: The theoretical basis of the article is ample, but research on the application of therapeutic drugs can be further supplemented.

Response 1: Thanks for your comment! The information on application of therapeutic heterobifunctional molecules is limited in this review article because current research has focussed on the proof-of-principle. The next logical step will be to translate these concepts to a potential clinical application. As of now, this has only been investigated for PROTACs and molecular glues, and is summarized in other reviews (such as DOI: 10.1038/s41573-021-00371-6).   

Point 2: The research on whether there is a negative side to the normal function of Heterofunctional Small Molecule In conductors should be supplemented in the original text.

Response 2: There are few studies on negative aspects of heterobifunctional molecules to date because, like mentioned above, research is in a fundamental phase. Many problems will probably not be discovered until preclinical and clinical studies begin. But for now, only PROTACs have made it into clinical trials. Nevertheless, I added a paragraph on PROTACs discussing their current problems and limitations (section 2.1.). These challenges hold true for most heterobifunctional molecules. Supplementing them has significantly strengthened the manuscript. Thank you!  

Reviewer 2 Report

The review described targeted modifications using a similar way to PROTAC. It would be an informative review and might be implemented toward the understanding of the functioning of phosphoproteome biology in the future.

There is some typo(s) in line 66 & 67.

Figure 2 needs more clarity, the author might consider adding figure legends for the symbols (like effector and POI are bridging but what in there in bait and prey format connected to linker) used in the figure for making it more clear and more understandable to the readers by looking at the figure only, although author has explained it in the text.

Typo(s) in line 141

Typo (s) in line 549..phosphorylation-targeting chimeras (PhoRCs), it's targeting or recruiting?

Author Response

Response to Reviewer 2 Comments

The review described targeted modifications using a similar way to PROTAC. It would be an informative review and might be implemented toward the understanding of the functioning of phosphoproteome biology in the future.

Point 1: There is some typo(s) in line 66 & 67.

Response 1: Revised. Thanks for pointing out the problem!

Point 2: Figure 2 needs more clarity, the author might consider adding figure legends for the symbols (like effector and POI are bridging but what in there in bait and prey format connected to linker) used in the figure for making it more clear and more understandable to the readers by looking at the figure only, although author has explained it in the text.

Response 2: Thanks for the suggestion! I’ve added a legend explaining the symbols for POI, effector protein and heterobifunctional chimera to enhance clarity for the reader. 

Point 3: Typo(s) in line 141

Response 3: Typo “öparts” has been replaced by “parts”. 

Point 4: Typo (s) in line 549..phosphorylation-targeting chimeras (PhoRCs), it's targeting or recruiting?.

Response 4: Typo has been corrected. It should read phosphatase-recruiting chimeras. Thank you! 

Reviewer 3 Report

This review nicely summarized all kinds of TAC strategies to manipulate (De-)ubiquitination, (De-)phosphorylation, acetylation and autophagy. The whole review is well organized and reads very well. Therefore, the manuscript can be accepted for publication in the journal with a minor revision. Below are some specific comments:

1.       In line 66, there is a German sentence needs to be deleted.

2.       In line 480, it should be University of California, Berkeley

3.       A few interesting and newly updated literatures can be added into this review: 1) trivalent PROTAC (PMID: 34675414); 2) radiotherapy-triggered PROTAC (PMID: 36542856); 3) HEMTACs (PMID: 36574496); 4) TF-DUBTAC (PMID: 35786952); 5) Targeted degradation via direct 26S proteasome recruitment (PMID: 36577875)

Author Response

Response to Reviewer 3 Comments

This review nicely summarized all kinds of TAC strategies to manipulate (De-)ubiquitination, (De-)phosphorylation, acetylation and autophagy. The whole review is well organized and reads very well. Therefore, the manuscript can be accepted for publication in the journal with a minor revision. Below are some specific comments:

Point 1: In line 66, there is a German sentence needs to be deleted.

Response 1: Revised. Thanks for pointing out the problem!

Point 2: In line 480, it should be University of California, Berkeley

Response 2: Revised.

Point 3: A few interesting and newly updated literatures can be added into this review: 1) trivalent PROTAC (PMID: 34675414); 2) radiotherapy-triggered PROTAC (PMID: 36542856); 3) HEMTACs (PMID: 36574496); 4) TF-DUBTAC (PMID: 35786952); 5) Targeted degradation via direct 26S proteasome recruitment (PMID: 36577875)

Response 3: Thanks for drawing my attention to these papers! They are excellent and very interesting works. Radiotherapy-triggered PROTACs, HEMTACs and direct proteasome recruitment have not been considered for this review in the first place due to the simple fact they had not been published at the time of submission. Short paragraphs were added dealing with radiotherapy-triggered PROTACs (chapter 2.1.1) and TF-DUBTACs (chapter 3).   
